# Membrane Interference Against HIV-1 by Intrinsic Antiviral Factors: The Case of IFITMs

**DOI:** 10.3390/cells10051171

**Published:** 2021-05-11

**Authors:** Federico Marziali, Andrea Cimarelli

**Affiliations:** Centre International de Recherche en Infectiologie (CIRI), Université de Lyon, Inserm U1111, CNRS, UMR5308, ENS de Lyon, Université Claude Bernard Lyon 1, 46 Allée d’Italie, 69007 Lyon, France

**Keywords:** HIV-1, IFITM, interferon, membrane, infection

## Abstract

HIV-1 is a complex retrovirus that is adapted to replicate in cells of the immune system. To do so, HIV-1, like other viruses, developed strategies to use several cellular processes to its advantage, but had also to come to terms with an arsenal of cellular innate defense proteins, or antiviral factors, that target more or less efficiently, virtually every step of the virus replicative cycle. Among antiviral restriction factors, the family of interferon-induced transmembrane proteins (IFITMs) has emerged as a crucial component of cellular innate defenses for their ability to interfere with both early and late phases of viral replication by inhibiting cellular and viral membranes fusion. Here, we review the enormous advances made since the discovery of IFITMs as interferon-regulated genes more than thirty years ago, with a particular focus on HIV-1 and on the elements that modulate its susceptibility or resistance towards members of this family. Given the recent advances of the field in the elucidation of the mechanism of IFITM inhibition and on the mechanism(s) of viral resistance, we expect that future years will bring novel insights into the definition of the multiple facets of IFITMs and on their possible use for novel therapeutical approaches.

## 1. Introduction

Intracellular parasites viruses rely on the cellular machinery to efficiently replicate within their host, a feature that obliges them to invariably come to terms and sometimes overcome what can be a hostile environment. Particular interest has been devoted to the study of innate defense factors and specifically to antiviral effectors of the type I interferon response (IFN-I). A number of exhaustive reviews on host proteins modulating HIV replication have been published elsewhere [1,2,3]. Here, we will focus on the interferon-induced transmembrane proteins (IFITMs), a unique family of restriction factors with a broad spectrum of viral inhibition and we will more specifically detail their relationship with HIV-1 as a paradigm for other viruses.

IFITMs belong to the dispanin/CD225 family that originated from metazoan lineages and diverged into four subfamilies (A through D) [4]. IFITMs are members of the A subfamily that in humans are part of a single locus on chromosome 11 (Figure 1a). IFITM1, 2 and 3, are IFN-induced thanks to the presence of an interferon-stimulated response element in their promoters [5,6,7] and are essentially studied in the context of viral infection; IFITM5 has been genetically linked to osteogenesis imperfecta type V, a bone-related disease [8] and IFITM10 whose functions remain unknown. Other members of the remaining dispanin/CD225 subfamilies have been either shown to be involved in vesicle trafficking (TUSC5/TRARG1, PRRT2, and TMEM90B/Syndig1; trafficking regulator of GLUT4, proline rich transmembrane protein 2 and synapse-differentiation inducing 1, respectively), or have no clear ascribed functions (PRRT1, TMEM233, TMEM90A, and TMEM91) [9].

IFITMs share a similar structure and present a hydrophobic intra-membrane-associated domain (IMD) that previous studies defined as transmembrane, a cytoplasmic intracellular loop (CIL), and a transmembrane domain (TMD), as well as N- and C-termini of variable lengths [11,12] (Figure 1a). By virtue of this organization, IFITMs are membrane-resident proteins and despite their high similarity, do present distinct intracellular distributions at steady state that are regulated by their Nter and Cter regions. As such, while IFITM1 is mostly localized at the plasma membrane, the distribution of IFITM2/3 is skewed towards membranes of endo-lysosomal compartments (Figure 1a).

## 2. An Historical Overview of IFITMs

IFITMs were discovered in the 1980s as the first transmembrane protein-coding genes whose expression was highly inducible by interferon alpha treatment in TG98 neuroblastoma cells (named at the time: 9-27, 1-8D and 1-8U) [5]. The first studies in knock-out mice suggested that IFITMs could play a role in the migration of primordial germ cells (PGC) and in germ cell development [13]. However, the relevance of such observations was later challenged by a report indicating that *ifitm3^-/-^* or *ifitmdel^-/-^* knockout mice (in which either *ifitm3* or the entire *ifitm* locus were ablated) exhibited no germ line developmental defects [14]. In the following years, increased levels of IFITMs expression were often associated to the status of cancer progression in different types of tumors (gastric, bladder, breast, colorectal, as well as acute myeloid and lymphocytic leukemias [15,16,17,18,19,20,21,22,23,24]). Given the well-established links between inflammatory responses and cancer development, it is not surprising that IFITMs appear as associated markers of this process. However, emerging evidence indicate that IFITMs may play a more active role in the tumorigenic process by acting as scaffolds for oncogenic signaling pathways like TGF-ß, Wnt/ß catenin, as well as the IGF1/IGF1R and PI3K/Akt/mTORC axes (see [25] for a review covering the relationship between IFITMs and cancer).

The first glimpse of the antiviral capacities of IFITMs was obtained in experiments showing the restriction of vesicular stomatitis virus (VSV) replication upon IFITM1 overexpression [26]. However, it was not until 2009 that IFITMs (and more specifically IFITM3) came back to the limelight as innate immune factors capable of inhibiting several viruses, following genome-wide shRNA screens for cellular modulators of the infection of influenza A (IAV), West Nile, and dengue viruses [27,28]. Ever since, a large number of studies by different laboratories have contributed to establish IFITMs as broad antiviral inhibitors capable of interfering with the replication of a very large list of DNA and RNA viruses derived from different families and among them the human immunodeficiency type 1 virus (HIV-1) and more generally primate lentiviruses (reviewed in [11,29,30,31]).

## 3. IFITMs Inhibition of HIV-1

In the case of HIV-1, the first evidence of antiviral effects of IFITM proteins came from work of the Liang laboratory [32] that described how the pool of IFITM proteins in target cells protected them from infection, in line with the most commonly described mechanism of viral inhibition for IFITMs. A few years later, ours and the Schwartz’s laboratories independently described a second mechanism of inhibition according to which the pool of IFITM proteins in virion-producing cells led to the de novo production of HIV-1 particles of decreased infectivity, property that we refer to as negative imprinting of virion particles infectivity [10,33]. Our laboratory then demonstrated that this property was also conserved against other viruses [34], highlighting IFITMs as a paradigm innate defense factors capable of inhibiting viruses at two distinct moments of their life cycle: during entry into target cells and during the production of novel virion particles from infected cells (Figure 1b). A third mechanism of HIV-1 inhibition has been reported more recently, based on which IFITMs can also interfere with HIV-1 protein translation [35]. How this mechanism relates to the action of IFITMs as membrane fusion inhibitors remains unclear.

### 3.1. Target Cell Protection

According to this mechanism of inhibition IFITMs meet incoming virion particles in endosomes and prevent the fusion between the viral and the cellular membranes, ultimately leading to virions degradation (Figure 1b). While this mechanism of inhibition was described already in 2013 [36,37], it was only recently that elegant studies imaged this process in real time [38,39], showing that virion particles in endosomes decorated with IFITMs exhibit a delayed kinetic of fusion between the viral membrane and the endosomal one.

Endosomes continuously fuse with lysosomes according to a kiss-and-run model, whereby the two vesicles engage in repeated transient fusion events until complete fusion occurs with kinetics of the order of minutes [40]. In this context, the delay imparted by IFITMs clearly overlaps with the normal kinetics of fusion between endosomes and lysosomes, indicating that a major function of IFITMs is to postpone fusion of viral membranes until endosomes containing the virus meet with lysosomes. These results are therefore of interest, not just because they place IFITM-mediated inhibition in the larger context of dynamic vesicular trafficking, but also because they suggest that potentiation of the antiviral effects of IFITMs may be achieved through compounds that increase endosomal to lysosomal transfer or that, similarly to IFITMs, act on membrane fluidity and kinetically slow viral to cellular membrane fusion so that the timing of this process overlaps with the one of endo-lysosomes.

When considered from a wider perspective, HIV-1 inhibition by IFITMs during the phase of target cell infection is intriguing.

Viral entry can be broadly classified according to the dependence of this process from pH, as pH-dependent and pH-independent [41]. In the first, passage through the acidic pH of endosomes is required to trigger a conformational change of viral glycoproteins that become competent for membrane fusion. In this respect, the entry of a large spectrum of viruses susceptible to IFITMs is pH-dependent (influenza virus, etc.) and takes place as expected in endosomes that contain both the virus and IFITMs. However, HIV-1 entry into target cells has been classically defined as a pH-independent process and the fusion between viral glycoproteins and cellular receptors occurs at the plasma membrane [42]. Yet, most studies concur in a strong HIV-1 inhibition also by IFITM2 and IFITM3, despite the fact that they are essentially distributed in endosomes [10,32,33,43,44,45].

While it is true that a fraction of these more internal IFITMs is nonetheless also present at the plasma membrane and that the dynamic distribution of IFITMs is likely to continuously recycle them to and from the plasma membrane facilitating the encounter with a virus that fuses at the plasma membrane, a second non-mutually exclusive explanation of this apparent conundrum may be the observation that HIV-1 can borrow a functional endosomal entry pathway [46,47,48,49]. This finding remains debated in light of the classical view of HIV-1 membrane fusion at the plasma membrane [50,51,52]. However, the possibility that incoming viruses may also enter cells through endosomal vesicles without the need for an acidic pH would explain the susceptibility of HIV-1 to IFITM1, 2 and 3 despite differences in their intracellular distributions.

A last important issue to consider in our view is that imaging studies indicated that Lassa virus particles, an IFITM-resistant virus member of the *Arenaviridae* family [38], seem to be excluded from IFITM3-containing vesicles. While these observations provide a simple explanation as to the IFITM-resistance of the Lassa virus, they raise a fundamental question as to how this may be possible. Lassa virus has been described to use late rather than early endosomes to access the cell cytoplasm [53,54] and a predominant distribution of IFITMs in the latter could potentially explain the absence of IFITMs in Lassa virus-containing vesicles. However, IFITM2/3 appear equally distributed between late and early endosomes [55]. As such, these results raise a fundamental question that remains to be addressed: are endosomes more heterogeneous than expected and IFITMs are specific markers of a yet uncharacterized endosomal subpopulation, or else are IFITMs driving specific changes in endosomes in which they are embedded that endows them with specific features?

The answers to these questions appear of key importance and provide yet another example of how the field of HIV and of virology more generally can shed light on our fundamental understanding of basic cellular processes.

### 3.2. Negative Imprinting of Virions Particles Infectivity: The Production of Virion Particles of Decreased Infectivity

Our laboratory reported, together with the Schwartz’s lab in 2014 and the Liu one in 2015 [56], that the expression of IFITMs in HIV-1-producing cells led to the production of virion particles of decreased infectivity, property that we refer to as negative imprinting of virion particles infectivity, because the marks of the effects of IFITMs on the virion progeny are apparent at the following cycle of infection [10,33]. Close inspection of virion particles indicated no major defects in the composition of viral gag and envelope structural proteins, albeit some studies have reported this (see relevant section below). In virion-producing cells, IFITMs coalesce with budding virions and are incorporated into virion particles that exhibit decreased infectivity due to their lower propensity to undergo membrane fusion, in line with the major functions of IFITMs as membrane fusion inhibitors (Figure 1b). Given that incorporation into virions occurs in particles of different viruses and is proportional to their intracellular levels of expression [34], we believe that the most likely mechanism of IFITM packaging into virions is a passive incorporation that is a consequence of their membrane distribution.

At present, it has been impossible to clearly separate the physical incorporation of IFITMs into virions from their antiviral effect despite intensive mutagenesis, essentially because it is not possible to generate physiologically relevant IFITM mutants that have lost membrane localization [45]. Hence, it remains to be formally demonstrated whether the physical presence of IFITMs into virion particles is required to alter infectivity, or else whether IFITMs can drive changes in viral membranes during the process of virion assembly independently of their physical incorporation. This issue is important because it could shed light on alternative mechanisms of action by IFITMs and perhaps indicate novel mechanisms through which IFITMs could influence the protein and/or lipid composition of viral membranes.

As it will be discussed in the following section, if most studies have focused on the biophysical changes that characterize IFITM-decorated membranes in target cells, none has addressed the behavior of virion particle membranes that have been generated in the presence of IFITMs to determine whether one or two distinct mechanisms are at play during target cell protection versus negative imprinting of virion particles infectivity.

### 3.3. Translation Inhibition of HIV-1

High expression of IFITMs has been recently reported to lead to a decreased rate of HIV-1 protein synthesis. This effect is dose dependent and affects protein translation from both spliced and non-spliced viral RNAs, by promoting the exclusion of viral RNA from polysomes [35]. Interestingly, the authors reported that the HIV-1 Nef protein can help relieve such defect. At present, it is unclear how this third property relates to the membrane fusion inhibitory properties ascribed to IFITMs and whether translation inhibition can also broadly affect other viruses. It is, however, of interest to note that glycoGag, a non-structural gag isoform produced by the murine leukemia virus (MLV), is also able to confer resistance to IFITM3 [57]. MLV glycoGag and HIV-1 Nef have been reported to counteract the restriction mediated by members of the serine incorporator (SERINC) family against HIV [58,59] and their apparently similar relieving effects against IFITM inhibition is of interest, despite the fact that IFITM3 resistance in the case of glycoGag does not seem to involve modifications in the steady-state levels of Env [57].

## 4. Molecular Basis of IFITMs Inhibition

Despite the fact that membrane fusion inhibition was rapidly recognized as the main mechanism of IFITM inhibition, the underlying mechanism has been the subject of intense studies with a number of non-mutually exclusive models that have been proposed over the years by different laboratories. As of today, the most accepted mechanism of membrane fusion inhibition is based on the direct rigidification of membranes by IFITMs themselves. However, alternative mechanisms have been proposed that, although proven not completely correct, may contribute to shed light on additional functions of IFITMs. As such, these mechanisms will also be discussed in this section.

### 4.1. Direct Membrane Rigidification

Membranes of cells expressing IFITM proteins are more rigid and this has been determined upon Laurdan staining coupled with two-photon and fluorescence-lifetime imaging microscopy (Laurdan is a hydrophobic fluorescent probe sensitive to lipid phases), as well as after use of a novel fluorescent lipid tension FliptR reporter [36,37,55,60,61]. In agreement with a model in which IFITMs induce more rigid membranes, amphotericin B, an antifungal antibiotic that instead fluidifies them, has been shown to oppose the effects of IFITM3 against different viruses [38,55,62].

Extensive literature exists on the effects that changes in the composition of lipid membranes as well as on the consequences that a more rigid membrane can bear for viral infectivity (for an introductory review [63]), so that this model can easily explain how IFITMs can drive membrane fusion inhibition.

At present, several lines of evidence argue in favor of a direct physical action of IFITMs on membrane rigidity: (i) the restriction of incoming viruses requires the colocalization with IFITM-decorated endosomes [38,39,44]; (ii) IFITM oligomerization is required for membrane rigidification and this correlates with viral restriction [55,64]; (iii) IFITM3 induces negative curvature in vitro which is consistent with membrane rigidification [60].

This model has however been proven exclusively for target cell protection and it is unknown whether it applies also to the negative imprinting phenotype of IFITMs in which there is no strict correlation between antiviral effects on virion particles and levels of IFITM incorporated into virion particles when a large panel of mutants are analyzed.

### 4.2. Indirect Biophysical Changes Due to Lipid Alteration

Paradoxically, the same experiments on the ordered phase of lipids mentioned above lend ground to the hypothesis of an indirect effect of IFITMs on the alterations of biophysical properties of membranes. Indeed, partly due to technical challenges in measuring Laurdan changes in endosomal membranes, increased lipid order has been essentially measured at the plasma membrane [36,55]. According to our data, the amount of IFITM2/3 at the plasma membrane in different cell types is at least ten-fold lower than it is at internal membranes [34], suggesting either that very low levels of IFITM3 are sufficient to durably change the biophysical properties of membranes, or else that the lipid phase ordering effects measured at the plasma membrane are the additive result of both a direct and an indirect effect of IFITMs. According to the first possibility, the lipid order in endosomes enriched with IFITM3 would be expected to be much higher than the one measured at the plasma membrane. The latter possibility would be of interest because it would allow IFITMs to commandeer wider and more global changes of cellular membranes.

IFITM3 has been reported to increase intracellular cholesterol levels by interfering with the transporter activity of the vesicle-associated membrane protein-associated protein A (VAPA) [65]. Although a number of reports could not provide supporting evidence for cholesterol changes driven by IFITM3 [37,45,66], the interaction between IFITM3 and VAPA, if confirmed, may bear implications that extend beyond cholesterol levels.

VAPA is an endoplasmic reticulum (ER)-resident general adaptor molecule that serves as a docking site for the recruitment of lipid transfer proteins at membrane contact sites (MCS) between the ER and the Golgi [67]. The VAPA-binding proteins described so far are the ceramide transfer protein (CERT) involved in ceramide transport towards the Golgi, the oxysterol-binding protein (OSBP) that concomitantly transfers cholesterol and PI4P to and from the Golgi, the four-phosphate adaptor protein 2 (FAPP2) that delivers glucosylceramides to the trans-Golgi network, as well as the phosphatidylinositol transfer protein, cytoplasmic 1 (PITPNC1) that mobilizes phosphatidic acid. The coordinated action of these proteins is important to maintain lipids and phospholipids fluxes at the ER-Golgi interface, influencing directly or indirectly the lipid composition of all cellular membranes [67,68,69,70,71]. It remains therefore possible that the interaction between VAPA and IFITM3 does not lead to drastic changes in cholesterol levels, but to more subtle ones in different classes of phospholipids. In light of the numerous effects that phospholipids play in several physiological processes including viral infection, this hypothesis deserves further studies in the future.

### 4.3. Env-Mediated Trafficking/Processing Defects

The HIV-1 glycoprotein is synthesized as a single polyprotein of 160 kDa (gp160) that enters the ER-Golgi secretory pathway and is processed by Furin in its mature forms: gp120 that remains at the surface of the virus particles and gp41, the transmembrane protein that tethers gp120 to the particle [42]. A few studies have reported decreased envelope expression and lower virion incorporation upon IFITMs overexpression in virion-producing cells [56,57] and while this mechanism of interference cannot of course explain target cell protection, it could potentially explain the lower infectivity of virions particles produced in the presence of IFITMs. Despite the fact that this decrease was initially thought to be mediated by a specific interaction between the cytoplasmic tail of HIV-1 gp41 and IFITM3 [56], subsequent studies reported similar effects on the murine leukemia virus (MLV) and vesicular stomatitis virus (VSV) envelope glycoproteins that do not possess such sequences, indicating that these effects are not virus specific and suggesting that IFITM3 could interfere with the normal processing route of several glycoproteins [57]. In support of this hypothesis, a population of immature HIV-1 Env precursors has been recently identified that reaches the plasma membrane bypassing the Golgi pathway and that, interestingly, is selectively excluded from virion particles [72]. Despite the fact that such alternative pathway remains to be fully characterized, this finding raises the possibility that IFITMs may decrease the amount of packageable Env trimers by skewing glycoprotein trafficking towards a non-productive pathway, similarly to the one described above.

Few considerations contribute however to tone down Env downregulation as the key mechanism of IFITM inhibition during the negative imprinting of virion particles infectivity. First, HIV-1 strains exhibit extensive differences in the amounts of envelope glycoproteins incorporated at the virion surface and the antiviral effects of IFITMs do not correlate with this parameter [73]. Second, in conditions in which the expression levels of IFITMs approximate what can be measured in primary cells stimulated with type 1 interferons, decreased Env glycoproteins incorporation is not observed for HIV-1, nor for a number of diverse viruses [33,34,44]. Third, virion particles produced in the presence of IFITMs do not display obvious defects in their ability to dock target cells, which would be instead expected in case of elevated gp120 shedding from particles [10].

Overall, while these results suggest that envelope downregulation is not the major mechanism of inhibition through which IFITMs decrease virions infectivity, they point nonetheless to the possibility that IFITMs may in some circumstances interfere with the glycoprotein secretory pathway in manners that remain to be elucidated.

## 5. HIV-1 Resistance towards IFITMs

Many of the viruses studied so far with respect to IFITMs present a binary trait, in that they are either susceptible or resistant to these antiviral factors. For the moment, HIV-1 has provided the sole example of distinct strain-specific behavior towards IFITMs. If initial studies indicated laboratory-adapted HIV-1 strains (for example, NL4-3) as susceptible to IFITMs, subsequent results from the Neil’s lab [44], later confirmed by numerous laboratories, revealed that transmitted founder HIV-1 isolates were instead largely IFITM-resistant.

HIV-1 transmitted founders are viruses isolated very early after infection of a patient, prior to the viral quasi-species expansion that occurs during the chronic phase. These strains exhibit unique features when compared to isolates derived from the chronic phase and in particular, a higher resistance to type I interferons (IFN-I) that can be partly ascribed to IFITM-resistance [44,74]. The major viral genetic determinant of this resistance has been mapped to the envelope glycoprotein [34,44,75,76] and specifically to several variable loops present in gp120: V3, the first identified, as well as V1 and V2 identified more recently [73].

The finding that Env is a genetic determinant for this distinct HIV-1 behavior is at odds with the observation that the genetic swapping of envelope glycoproteins from resistant viruses (hepatitis C virus, HCV, and Rift Valley fever virus, RVFV) to IFITM-susceptible viruses (HIV-1 and VSV) does not transfer resistance. While these results indicate that envelope glycoproteins are not sufficient alone to transfer resistance between distant viruses, the results obtained in the case of HIV-1 clearly indicate that this is the case within an homologous situation in agreement with the observation that adaptive mutations in Env appear upon prolonged passage of HIV-1 in cells expressing IFITMs [56]. To date, there is no evidence for a direct antagonism between the HIV-1 Env and IFITMs in that IFITMs do not appear to be degraded, nor mislocalized during infection, as it is the case for other restriction factors (for example, Tetherin or SERINC5). Hence, the most likely explanation is that mutations in Env result in molecules able to complete their tasks within the constraints of a more rigid viral membrane, as is the one imposed by IFITMs.

HIV-1 envelopes are very complex and heterogeneous proteins. First, the number of Env trimers at the surface of virion particles varies considerably among viral isolates and this parameter does not strictly correlate with virion infectivity (i.e., virions with more trimers are not necessarily more infectious [73,77,78,79]). Second, as estimated by single-molecule Forster resonance energy transfer (smFRET), these trimers can exist in three conformations: a closed, pretriggered conformation, an open fully CD4-engaged conformation, and an intermediate one (defined as state 1, state 3, and state 2, respectively) [80,81]. Third, Env proteins differ also with respect to the co-receptor they engage, which can be either the CXCR4 and/or the CCR5 chemokine receptors (viral isolates that use one or the other are referred to as X4 and R5-tropic, respectively).

In general, HIV-1 strains most affected by IFITMs present envelopes with lower gp120/gp41 association indexes and are more easily neutralized by soluble CD4 [75]. Such envelopes are hypothesized to sample more frequently open conformations and to display higher energetic state and lower conformational stability [73,76] that IFITMs can further exacerbate. It is of interest that lab-adapted strains susceptible to IFITMs are X4-tropic, while resistant ones are R5-tropic and that the V3 loop that confers resistance to IFITMs specifies R5-tropism and contains less polar residues that are thought to promote close envelope conformations. However, a very recent study compared a large selection of X4- and R5-tropic transmitted founder viruses and failed to reveal a clear correlation between behavior towards IFITMs and either virus tropism or quantities of Env trimers present at the virus surface [73].

After receptor/co-receptor binding, several Env trimers cluster (i.e., move laterally in viral membranes), similarly to cellular receptors on cellular membranes for efficient membrane fusion to occur (Figure 2) [82,83]. A plausible working model would be that IFITM-resistant Env exhibit higher affinities for their receptors and hence require the lateral displacement of fewer trimers to trigger fusion. Alternatively, IFITMs can alter the conformation of Env trimers so that only most stable ones remain functional in the presence of IFITMs. Several studies have indicated that the conformation of gp120 is particularly sensitive to changes that occur at the other side of the viral membrane, in the interior of the virion particle. Indeed, gp120 is anchored to the virion membrane by gp41, that is in turn in contact with the internal structure of the virus via an unusually long cytoplasmic tail accommodated during the process of virion assembly into trimers formed by the matrix domain (MA) of gag. A few studies have indeed determined that the maturation status of gag and therefore the manner in which the cytoplasmic tail of gp41 interacts with the MA trimeric structure can influence the conformation of gp120 [83,84,85,86,87,88]. Since IFITMs are posited in viral membranes, they may therefore exert tensions along this axis to influence the overall conformation of Env trimers.

## 6. Final Considerations and Perspectives

In conclusion, the mechanism of IFITMs inhibition is likely linked to the stoichiometry and of conformation of viral and cellular proteins involved in viral entry. The exact mechanism(s) through which particular HIV-1 Env proteins are endowed with the ability to cope with the more ordered membrane environment specified by IFITMs remains unclear and future studies will be required to elucidate this issue.

Similarly, while HIV-1 represents, for the moment, the only example of a virus for which resistance and susceptibility to IFITMs can be observed according to the viral strain examined, it is unknown whether such behavior may be common to other viruses, as most studies have focused on one single viral strain. This topic needs to be explored more thoroughly, as this may underlie the importance that dynamic viral adaptation to IFITMs may play during viral replication and pathogenesis, as seems to be the case for HIV-1.

Another point of interest will be the study of IFITMs in the context of fully engaged IFN-I responses. For the moment, IFN-I are known to regulate IFITMs expression transcriptionally. However, IFN responses induce also a series of additional antiviral factors that may concur to the antiviral functions of IFITMs. For instance, IFN-I is a strong inducer of the cholesterol-25-hydroxylase (CH25H), an enzyme that converts cholesterol to 25-hydroxycholesterol and that also acts as a broad antiviral inhibitor of membrane fusion [89]. Further studies focused on the potential connections between IFITMs, CH25H, and potentially additional factors will certainly enrich our understanding of how IFITMs act in the context of IFN responses.

### Beyond Viral Inhibition: IFITMs as Double-Edge Swords?

Restriction factors protect cells from viral infection, yet, in some cases, their action can be deleterious for the cell itself. A good example of this is provided by members of the apolipoprotein B mRNA editing enzyme, catalytic polypeptide-like 3 (APOBEC3) family. APOBEC3s are cytidine deaminases that act as innate defenses against retroviruses and retro-elements, by extensively mutagenizing reverse transcribing viral genomes in a process that has been defined as death by mutagenesis. Certain APOBEC3 members can access the cell nucleus and are able to attack the cellular genome itself, contributing to the overall process of tumorigenesis (reviewed in [90]).

In this respect, IFITMs are broad membrane fusion inhibitors and this process underlies basically all membrane trafficking events from the fusion between vesicle to target soluble N-éthylmaleimide-sensitive-factor attachment protein receptor (v- and t-SNAREs) in the Golgi, to the production and uptake of exosomes during the normal process of cell-to-cell communication (reviewed in [91]).

It is therefore plausible to ask whether the antiviral activity of IFITMs comes with a cost for the cell. IFITMs have been shown to be involved in glucose metabolism, albeit through mechanisms that remain unclear for the moment [92]; they have been shown to participate in B cell signaling [93] and can also interfere with trophoblast fusion during placental formation [94,95]. In addition, we have discussed in this review reports suggesting that IFITMs may interfere more generally with the cellular secretory pathway [56,57] and perhaps even cellular translation [35].

While it is true that IFITMs are controlled transcriptionally and heavily regulated post translationally, their expression is often deregulated in cancer and is expected to be so in the large spectrum of pathologies regrouped under the name of interferonopathies. It is therefore a non-speculative question to ask how a general membrane fusion inhibitor can influence basic cellular process such as Golgi trafficking, vesicular fusion, and even exosome secretion and uptake. We hope that future studies will address some of these issues and in so doing enrich on one hand our current understanding of the biology of this family of antiviral factors and on the other our comprehension of fundamental cellular processes.

## Figures and Tables

**Figure 1 cells-10-01171-f001:**
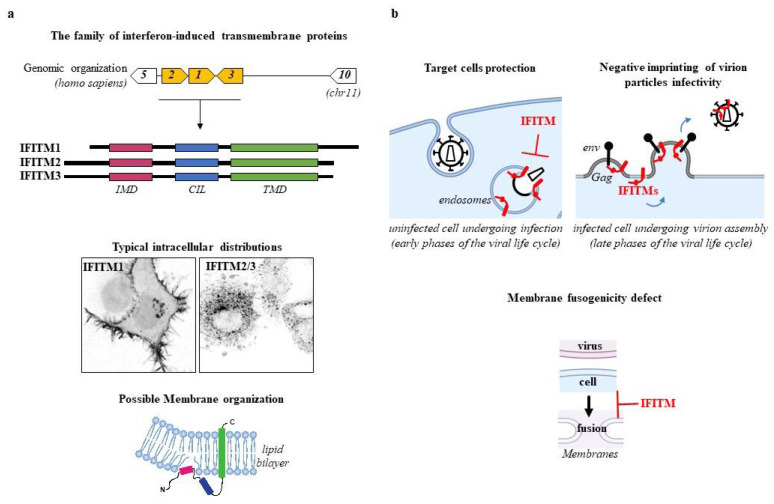
IFITMs and their effects against HIV-1. (**a**) Schematic presentation of the genomic organization of the IFITM locus on human chromosome 11 and focus on IFITM1, 2 and 3 that are IFN-regulated and studied in the context of viral infection. Typical confocal microscopy pictures presenting distribution differences between IFITM1, 2 and 3 (as published in [10]). Possible topological conformation of IFITMs on the membrane bilayer. (**b**) Schematic representation of the two mechanisms of HIV-1 interference by IFITMs on target cell protection (left) and negative imprinting of virion particles infectivity (right). In both cases, IFITMs interfere with the fusion between viral and cellular membranes impairing infection (bottom). Given that its relationship with membrane fusion inhibition as well as its possible conservation for other viruses remain unclear, the third mechanism of HIV-1 interference by IFITMs, namely protein translation inhibition, is not reported in the figure. Figures in section b were adapted from “Membrane Endocytocis” and “Hemifusion” templates by BioRender.com (2021), accessed on the 11th of May 2021).

**Figure 2 cells-10-01171-f002:**
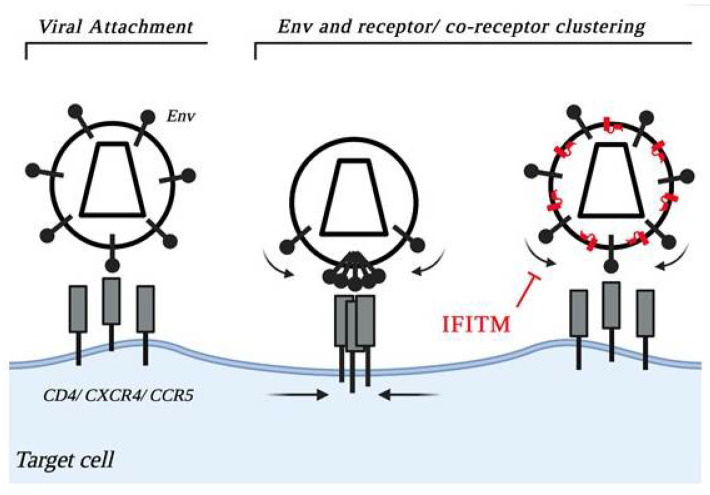
A possible mechanism of IFITMs inhibition during infection. After receptor/co-receptor binding, several Env trimers cluster for efficient membrane fusion to occur, similarly to cellular receptors on cellular membranes. This clustering event consists in the lateral displacement of several Env proteins through the lipid bilayer. By rigidifying their environment, IFITMs may interfere with the movements of Env molecules impeding clustering and therefore fusion. We hypothesize that a possible manner to circumvent this block is through Env proteins that display higher affinities for their receptor and that therefore require the lateral displacement of fewer Env trimers in order to start membrane fusion. Alternatively, through their action on viral membranes, IFITMs may perturb the overall structural conformation of the gp120-gp41-MA axis, an effect that would more drastically alter less stable gp120 trimers. For simplicity, the possible effects of IFITMs on clustering are depicted only in the case of the negative imprinting of virion particles infectivity, although the model can apply also to target cell protection. Alternative models of IFITMs inhibition are not presented in the figure.

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
