# Peer review of "Membrane Interference Against HIV-1 by Intrinsic Antiviral Factors: The Case of IFITMs"

_cells, 2021, doi:10.3390/cells10051171_

Round 1

Reviewer 1 Report

Here, the authors propose a review on the mechanisms of action of IFITMs restriction factors on HIV-1 at the membrane level. The Review is quite well written, the figures are clear, but there are several major points that merit discussion:

  • The article could be more easily readable with a classic layout such as 1/ mechanisms of action against viruses 2/ HIV and IFITM (at the membrane level) 3/ viral escape mechanisms
  • The mechanism of action of type I interferon on IFITMs is not fully described: expression increase? Clustering of IFITM?
  • In the introductory paragraph, IFITM are not presented as restriction factors
  • There is no description of the type of virus inhibition by IFITMs: RNA viruses? DNA viruses? HIV X4 and R5-tropic viruses. A table could be informative, together with the type of IFITM involved.
  • The authors quote that IFITM proteins inhibit HIV-1 protein synthesis, excluding viral mRNA from translation (Lee et al, line 402): this mechanism of action could be explained more precisely.
  • As a viral escape to the virus, two hypothesis are evoked (line 365): higher affinity of Env for their receptor, and structural conformation change of the gp120-gp41. Maybe another mechanism could be quoted: Env mutants that overcome IFITM restriction (Yu et al, Cell Rep 2015)

Minor points are also noted:

  • There is a repetition in the abstract (line 22). The abstract do not present IFITM as restriction factors.
  • Line 69: Interferon alpha
  • There is no mention of HIV-2 as a virus susceptible to IFITMs.
  • Line 397 : “IFITMs have been shown to be involved in glucose metabolism” : this could rather takes place in the introduction.
  • Therapeutic approaches announced in the abstract are not detailed in the manuscript.

Author Response

Please, find below the answers (blue) to the specific concerns expressed by the reviewer (black).

General remarks

The reviewer defines the review as quite well written and with clear figures, and highlights several points that deserve further discussion.

 We are pleased with the reviewer’s comments and in agreement with their suggestions we have adopted all of their suggestions (see below).

Major Issues

1. The reviewer suggests a more classical layout for the review.

  We have adopted what we believe a more readable chapter format for the review and we have maintained individual subchapters where this seemed more appropriate.

2. The reviewer suggests to better describe the mechanism of action of IFN-I  on IFITMs.

In this review we only detail the transcriptional regulation of IFITMs by IFN-I, which is to date their only known mechanism of regulation. However, we have now added an additional paragraph highlighting possible additional connections between the IFN response and IFITMs (lines 447-463).

3. The reviewer asks us to present IFITMs as restriction factors in the    introduction.

  Accordingly, we have clearly stated this in the abstract and introduction (lines 13 and 32).

4. The reviewer suggests to describe more extensively the viruses inhibited   by IFITMs perhaps adding a table.

 We have specified that IFITMs can block both RNA and DNA viruses (line 117). However, we feel that a table with the list of viruses inhibited by IFITMs would be distractive from the main focus of this review that is centered on HIV-1.

5. The reviewer asks us to describe more extensively the mechanism of HIV-1 translation inhibition by IFITMs.

  Accordingly, we have introduced a novel subchapter (lines 232-245).

6. The reviewer suggests to include Env adaptive mutations as a possible manner to overcome IFITM restriction.

 We now quote the work mentioned by the authors (lines 376-377). We feel that the presence of such Env mutants fits the proposed mechanisms of resistance to IFITMs already evoked in the relevant section.

Minor Issues

  1. We have deleted the paragraph that was inadvertently copied twice in the submitted version and we have presented IFITMs as restriction factors in the abstract.
  2. We have corrected interferon alpha in extenso (line 93, please note that there is a jump in lines number that we cannot fix from 42 to 70).
  3. We have mentioned the broad action of IFITMs on primate lentiviruses (line 120).
  4. We have preferred to keep the possible involvement of IFITMs in glucose metabolism in the subchapter that deals with their additional functions (from line 466).
  5. We agree that the therapeutic approaches are only hinted here in this review. But a hint on them is described in lines 158-162 .

Reviewer 2 Report

This review is well written. The topic of IFITMs activity against HIV is well described, taking into account the most recent knowledge and critically proposing different models of mechanisms of action. I do not have major criticism but only a few minor points.

1) figure 1: it would be useful to add some information about the panel showing the cellular distribution of IFITMs. If this panel is from a figure not published earlier, please give some information (are these cells transfected to over-express the indicated IFITMs? what cells are they? etc...). Otherwise, insert a reference.

2) The last paragraph of the abstract is duplicated

3) Line 97: the “most common mechanism” I believe is supposed to mean the “most commonly described mechanism”.

4) Line 165: to me, the meaning of this title is confusing. Perhaps a colon instead of “or” makes it clearer.

5) Line 297: Is the “binary trait” reported here real or is it a consequence of the fact that for viruses other than HIV different isolates were not tested?

6) Line 351 and 358: what does “the external conformation of gp120” mean? Is there an internal conformatiom of gp120?

7) Lines 369-370. The alternative model described in the figure legend is not depicted in the figure. Either state this in the figure legend or add the model to the figure.

Author Response

Please, find below the answers (blue) to the specific concerns expressed by the reviewer (black).

General remarks

The reviewer defines the review well written, with a well described topic and critical presentation of the different models of action of IFITMs.

 We are of course very pleased with the reviewer’s comments and in agreement with their suggestions we have adopted their suggestions (see below).

Minor Issues

  1. We have referenced the cellular distributions of IFITMs in Fig 1 to the published reference (line 77).
  2. We have deleted the paragraph that was inadvertently copied twice in the submitted version.
  3. We have corrected to the most commonly described mechanism (line 125).
  4. We have replaced or with a colon (line 202).
  5. We believe that the binary trait is a consequence of the fact that a single viral strain has been tested for the majority of other viruses. We extend on this in the final considerations chapter (lines 444-449).
  6. We have eliminated external from the conformation of gp120 (lines 412 and 420).
  7. We have stated that only one model is discussed and presented in the figure.

Reviewer 3 Report

Manuscript ID: cells-1197136

Title: Membrane interference against HIV-1 by intrinsic antiviral factors: the case of IFITMs

Comments to the Authors

In the review entitled “Membrane interference against HIV-1 by intrinsic antiviral factors: the case of IFITMs”, by Federico Marziali and Andrea Cimarelli, the authors review the studies relating to IFITMs with a particular focus on its effects on HIV-1.  

Major Comments:

  1. Although, the review is of interest to virologists and could also lead to a better understanding of HIV-1 infection and the use of IFITMs as therapeutic agents, clearly there are problems with the English language. The review needs to be thoroughly read and corrected by someone well versed in the English language. There are numerous examples throughout the manuscript that need to be corrected. I will provide two examples. Line 34 states, “..to the study of innate defense factors and specifically to actors of the type I interferon…” Actors is not the correct term to use here. Line 38, “transmembrane proteins, (IFITMs), a peculiar family of proteins with a broad…” “Peculiar” is not the correct term to use. A better word would probably be “unique.”
  2. In the abstract lines 22-27 are an exact repetition of lines 15-21.
  3. The section on “Beyond HIV-1: IFITMs as double-edge sword, ” starting at line 381 is very disjointed and doesn’t reallyrole of  talk about HIV-1 or address the title. The very first sentence makes no sense. The relationship between IFITMs and APOBEC family of proteins is not clear.
  4. The conclusion (lines 373-380) should be moved to the end. Again, the first sentence in the conclusion makes no sense.
  5. IFITM proteins can inhibit HIV-1 replication through three distinct processes: (i) inhibition of viral entry; (ii) reduction of viral particle infectivity (negative imprinting); and (iii) inhibition of viral protein synthesis. The authors have not discussed the third process. The role of IFITMs in blocking protein synthesis and how HIV Nef can help overcome IFITM-mediated restriction of virus production needs to be elaborated.

Author Response

Please, find below the answers (blue) to the specific concerns expressed by the reviewer (black).

General remarks

The reviewer defines the review of interest to virologists, but is very critical of the writing.

We are a bit surprised about this comment on a review that two other reviewers defined well written and quite well-written. We are also lukewarm about the “makes no sense” phrasing used by this reviewer for paragraphs that instead seem to make perfect sense to other referees. However, in keeping with the overall criticism expressed by the reviewer we have extensively edited the manuscript.

1. We have changed actors to antiviral effectors and peculiar to unique (lines 29 and 32) and edited the entire review.

2. We have deleted the paragraph that was inadvertently copied twice in the submitted version.

3. We have rephrased the title and first lines of the subchapter: beyond viral inhibition: IFITMs as double-edge swords? The chapter is meant to discuss the possibility that there may be a price to pay for protection from viral infection and that IFITMs may exert deleterious effects on the cell physiology, similarly to what observed for another well-known family of restriction factors (APOBEC3s), the expression of which has been linked to tumorigenesis.

4. We have introduced a Final Considerations chapter and rephrased accordingly (lines 440-444).

5. We have introduced protein translation as a third mechanisms of IFITM-driven inhibition of HIV-1 (lines 235-248).

Round 2

Reviewer 1 Report

The authors have responded to the queries asked. However therapeutics perspectives were not evoked in the final manuscript.

Author Response

Dear Reviewer,

 the therapeutic perspectives have been incorporated into a paragraph rather than on a separate chapter  (lines 158-162).

Reviewer 3 Report

The authors have extensively revised and edited the manuscript and added additional details. This has improved the manuscript considerably and the revised manuscript reads very well. The authors have addressed all of my concerns.

Author Response

We are pleased with the positive comments of the reviewer about our revised version.